# The Status and Risk Factors of Brucellosis in Smallholder Dairy Cattle in Selected Regions of Tanzania

**DOI:** 10.3390/vetsci10020155

**Published:** 2023-02-14

**Authors:** Isaac Joseph Mengele, Gabriel Mkilema Shirima, Shedrack Festo Bwatota, Shabani Kiyabo Motto, Barend Mark de Clare Bronsvoort, Daniel Mushumbusi Komwihangilo, Eliamoni Lyatuu, Elizabeth Anne Jessie Cook, Luis E. Hernandez-Castro

**Affiliations:** 1Department of Global Health and Biomedical Sciences, School of Life Science and Bioengineering, The Nelson Mandela African Institution of Science and Technology (NM-AIST), Arusha P.O. Box 447, Tanzania; 2Tanzania Veterinary Laboratory Agency (TVLA), Central Zone Laboratory, Dodoma P.O. Box 1752, Tanzania; 3International Livestock Research Institute (ILRI), P.O. Box 30709, Nairobi 00100, Kenya; 4Center for Tropical Livestock Genetics and Health (CTLGH), International Livestock Research Institute (ILRI), P.O. Box 30709, Nairobi 00100, Kenya; 5Tanzania Veterinary Laboratory Agency (TVLA), Central Veterinary Laboratory, Dar Es Salaam P.O. Box 9254, Tanzania; 6The Epidemiology, Economics and Risk Assessment (EERA) Group, The Roslin Institute at the Royal (Dick) School of Veterinary Studies, University of Edinburgh, Midlothian EH25 9RG, UK; 7Center for Tropical Livestock Genetics and Health (CTLGH), The Roslin Institute at the Royal (Dick) School of Veterinary Studies, University of Edinburgh, Midlothian EH25 9RG, UK; 8Tanzania Livestock Research Institute (TALIRI), Dodoma P.O. Box 834, Tanzania; 9International Livestock Research Institute (ILRI), Dar Es Salaam P.O. Box 34441, Tanzania

**Keywords:** brucellosis, dairy cattle, cELISA, seroprevalence, risk factors, control

## Abstract

**Simple Summary:**

Brucellosis is a neglected, bacterial zoonotic disease that affects domesticated animals and people. Infection in cattle is caused by *Brucella abortus* which causes nonspecific clinical signs in female cattle including lowered milk production, infertility, and abortion in the last trimester. To determine the prevalence and the risk factors associated with brucellosis exposure, we conducted a cross-sectional study of smallholder dairy cattle in six regions of Tanzania, between July 2019 and October 2020. A total of 2048 dairy cattle blood samples were collected and tested for the presence of anti-*Brucella* antibodies. An overall seroprevalence of 2.39% was found with the highest seroprevalence in the Njombe Region (15.5%). The risk factors that were identified to be significantly associated with brucellosis seropositivity were having goats around dairy cattle and a history of abortion within a farm. The study findings suggest that brucellosis is still present in smallholder dairy cattle at varying levels among the studied regions. Education of smallholder dairy keepers is required regarding the disease, as well as risk and control measures for the disease. A One Health approach is required to study the role of small ruminants in the spread of the disease and to evaluate the public health risk to smallholder dairy farmers, especially in the Njombe Region.

**Abstract:**

Bovine brucellosis is a bacterial zoonoses caused by *Brucella abortus*. We conducted a cross-sectional study to determine brucellosis seroprevalence and risk factors among smallholder dairy cattle across six regions in Tanzania. We sampled 2048 dairy cattle on 1374 farms between July 2019 and October 2020. Sera were tested for the presence of anti-*Brucella* antibodies using a competitive enzyme-linked immunosorbent assay. Seroprevalence was calculated at different administrative scales, and spatial tests were used to detect disease hotspots. A generalized mixed-effects regression model was built to explore the relationships among *Brucella* serostatus, animals, and farm management factors. Seroprevalence was 2.39% (49/2048 cattle, 95% CI 1.7–3.1) across the study area and the Njombe Region represented the highest percentage with 15.5% (95% CI 11.0–22.0). Moreover, hotspots were detected in the Njombe and Kilimanjaro Regions. Mixed-effects models showed that having goats (OR 3.02, 95% C 1.22–7.46) and abortion history (OR 4.91, 95% CI 1.43–16.9) were significant risk factors for brucellosis. Education of dairy farmers regarding the clinical signs, transmission routes, and control measures for brucellosis is advised. A One Health approach is required to study the role of small ruminants in cattle brucellosis and the status of brucellosis in dairy farmers in the Njombe and Kilimanjaro Regions.

## 1. Introduction

Brucellosis is a global, neglected bacterial zoonosis caused by an intracellular, aerobic, Gram-negative, nonencapsulated, coccobacillus bacteria of the genus *Brucella* [1,2,3]. Among *Brucella* species, *B. abortus*, *B. melitensis,* and *B. suis* infect cattle, small ruminants, and pigs, respectively [4,5]. In cattle, however, infection with *B. melitensis* or *B. suis* can also occur depending on disease transmission dynamics such as regular contact between dairy cattle and small ruminants or pigs at a farm or on grazing land [4,6]. Cattle transmission occurs after ingesting *Brucella*-contaminated feed and water from the uterine discharges, abortion materials, or fetal membranes of infected animals [7]. Bovine brucellosis causes nonspecific signs such as abortion during the last trimester, retained placenta, reduced milk production, orchitis, epididymitis, and rarely, arthritis [4,8,9]. A number of predisposing factors have been associated with brucellosis seropositivity such as older age of animal, history of abortion, large herd size, access to surface water, location, and contact with other animals [10,11,12,13,14,15].

The Tanzanian cattle production system is divided into three main sectors (e.g., pastoral, agropastoral, and dairy), which are all similarly affected by brucellosis. For instance, brucellosis seroprevalence in cattle kept by pastoral communities has been reported to be as high as 30% [16,17], whereas in agropastoral communities, seroprevalence has been reported to be up to 11.3% [14,18,19,20]. Smallholder dairy cattle are commonly crossbreeds of indigenous cattle with exotic breeds such as Friesian, Ayrshire, or Jersey.

The first brucellosis outbreak occurred in imported dairy cattle in 1927 [21], and the first brucellosis diagnosis was made in 1928 by using a serum agglutination test [22]. It was not until the 1970s that the Tanzania government started a brucellosis control programme in dairy cattle, which lasted until late 1990s. During that time, active surveillance, testing and slaughter, and calf vaccination using the *B. abortus* Strain 19 (S19) vaccine, were implemented which led to a reduction in brucellosis prevalence from 15.2% [21] to 2% [16]; since then, there have not been similar control programmes implemented by the government. As a result, there has been a number of reports of an increased trend of brucellosis seroprevalence in several regions, reaching up to 21% [12,13,14,19,23,24,25]. Although several studies have evidenced an increase in brucellosis within different farming systems, the factors driving brucellosis transmission in dairy systems remain unclear. Therefore, our study aims to establish the status and risk factors associated with brucellosis seroprevalence in smallholder dairy farming systems in six regions of high milk production in Tanzania.

## 2. Materials and Methods

### 2.1. Study Area

This study was conducted from July 2019 to October 2020 across six administrative regions in Tanzania with the highest density of smallholder dairy cattle [26], i.e., the Arusha, Kilimanjaro, and Tanga Regions in the Northern Highland zone with 252,554 head of dairy cattle and the Iringa, Njombe, and Mbeya Regions in the Southern Highland zone with 103,306 head of dairy cattle (Figure 1). The 2019–2020 agricultural census reported that the nationwide top three regions with the highest number of dairy cattle (in decreasing order) were Tanga (>140,000), Arusha (>100,000), and Mbeya (>80,000). Other regions showed high densities of dairy cattle such as Kilimanjaro (>60,000), Iringa (>30,000), and Njombe (>20,000) [27].

### 2.2. Study Design and Sampling

Risk factors and the status of brucellosis at selected smallholder dairy farms in Tanzania were explored using a cross-sectional study. Most smallholder farms in these regions are characterized for having crossbred cattle of Friesian, Ayrshire, and Jersey with Tanzania Short Horn Zebu (TSHZ). However, Friesian crosses comprise the largest proportion (80%) of all breeds. Two main management systems are recognized in the area: (1) an intensive system in which pastures are cut and provided to livestock and (2) an extensive system in which cattle are left to graze in private or communal land.

The cattle in this study were selected from a subset of the dairy cattle registry of the Africa Dairy Genetics Gains (ADGG) program (https://data.ilri.org/portal/dataset/adgg-tanzania, accessed on 1 June 2019). The ADGG project has registered over 52,500 cattle across the regions from volunteer farmers, and approximately 4000 cattle were randomly selected for genotyping as part of genetic evaluations of this crossbred population [28]. The selected cattle had known genetic characteristics and could be identified by their preliminary information such as an ear tag number, age, and sex from the ADGG database [28].

These genotyped dairy cattle distributed across the 6 regions were the target of this study for establishing the disease phenotype. Due to small herd size in most farms, only one animal was genotyped, and therefore, sampled. However, the final sample size from each region was sufficient to estimate the seroprevalence of 5% (with 3% precision) and 95% confidence interval for the smallest region assuming simple random sampling [29,30].

### 2.3. Questionnaire Administration

A questionnaire tool was used to collect data on possible risk factors for brucellosis. The tool was developed and piloted before a final version was uploaded using the Open Data Kit (ODK) software platform on an International Livestock Research Institute (ILRI) server in Kenya (Appendix A). Then, the forms could be downloaded for completion on farm from the Google play app store onto an Android tablet (Samsung, Suwon-si, Republic of Korea).

Farmers or their representative were informed on the study aims and General Data Protection Regulation (GDPR) compliance. Farmers needed to sign a consent form before being interviewed in Kiswahili, and their cattle sampled. The questionnaire covered details of the respondent and their brucellosis knowledge, farm location, and herd management practices such as feeding and watering, disease control, abortion material handling, and cow or farm abortion history.

### 2.4. Blood Sampling, Pre-Analysis Processing, and Storage

Dairy cattle were manually restrained using ropes (halter) and hands, and 20 mL of blood was collected from the jugular vein using a sterile needle in two plain vacutainer tubes (BD Vacutainer^®^, Becton, Dickinson and Company, Franklin Lakes, NJ, USA). Tubes were barcoded (field barcode) and labeled with an animal identification number and collection date, and then stored in a cool box with ice packs before being transported to the Tanzania Veterinary Laboratory Agency (TVLA) zonal laboratories. Plasma and serum were left to separate overnight and tubes were centrifuged the next day at 3000 revolutions per minute for 5 min. Then, serum was collected and aliquoted into 2 mL cryovial tubes which were labeled and barcoded (laboratory barcode) as above before storing them at −20 °C. Tube barcodes, animal identification numbers, and collection date were recorded in a Microsoft Access^®^ database which was later linked to the ODK questionnaire metadata. Finally, cryovials were transported at controlled temperature to the Nelson Mandela African Institution of Science and Technology (NM-AIST) in Arusha, Tanzania for storage at −20 °C until serological analysis was conducted.

### 2.5. Serological Analysis

All samples were tested according to the manufacturer’s instructions using a competitive enzyme-linked immunosorbent assay (cELISA) (COMPELISA 160 & 400, APHA Scientific, Weybridge, UK). The World Organisation for Animal Health (WOAH) recommends the use of cELISA for confirming prevalence of infection to *Brucella* in animals, furthermore, the test has high sensitivity (97.9%) and specificity (~100%) and can be used for testing poor quality serum samples [31,32]. Briefly, the test serum from the freezer and all reagents from the refrigerator (except for conjugate stored in −20 °C) were thawed at room temperature. Each test serum (20 µL) was placed in wells from columns 1 to 10 and 6 wells were left for control positive (20 µL) serum, 6 wells for control negative (20 µL) serum, and 4 wells for (20 µL) conjugate control within columns 11–12. Immediately after reconstitution, 100 µL of the conjugate was added to all 96 wells.

Plates were covered with a lid, shaken vigorously (200 revs/min) in a microtiter plate shaker for 2 min to allow mixing, and incubated at room temperature (21 ± 6 °C) for 45 min on a rotary shaker at 160 revs/min. Plate contents were shaken out, plates were washed 5 times using washing solution under low pressure, and finally, dried by taping onto a layer of absorbent towels until no more liquid was removed.

Next, 100 µL of prepared o-phenylenediamine dihydrochloride (OPD)-chromogen/substrate solution was added to each well on each plate which was incubated at room temperature for 20 min. Then, 100 µL of stopping solution was added to all wells to stop further reactions. Plate well absorbance was measured at 450 nm within 10 min using a SYNERGY|HTX multi-mode reader (BioTek Instruments, Winooski, VT, USA).

#### 2.5.1. Plate Acceptance Criteria

The plate was considered to be valid when the mean optical density (OD) of the 6 negative control wells was greater than 0.700 (the optimal mean negative OD is 1.00), the mean OD of the 6 positive control wells was less than 0.100, the mean OD of the 4 conjugate control wells was greater than 0.700 (the optimal mean conjugate control wells is 1), and lastly, when the binding ratio (mean OD of negative control/mean OD of positive control) was greater than 10 [33,34].

#### 2.5.2. Test Interpretation

Lack of color development indicated that the tested serum sample was positive. A positive/negative cut-off value was calculated as 60% of the mean OD of the four conjugate control wells. Any test sample giving an OD equal to or below this value was regarded as being positive.

### 2.6. Data Management and Analysis

Questionnaire data were downloaded from ODK and the laboratory data from the Access database were imported into RStudio, https://www.r-studio.com (accessed on 10 January 2022) and were joined and cleaned before analysis.

Animal level seroprevalences were calculated as the proportion of seropositive animals divided by the total number of animals tested or by region for both overall and regional seroprevalences. In addition, an overall adjusted seroprevalence was also estimated with a 95% exact binomial confidence interval for the reported cattle populations in each region (stratum) using the *svydesign* and *svyciprop* functions of the survey R package [35]. The stratum level 95% confidence interval was calculated using the *binom.test* function from R package *stats*.

A spatial scan statistic was used to detect statistically significant spatial clusters of seropositive animals in the Njombe and Kilimanjaro Regions only. Cluster analyses were performed using the SaTScan™ v10.1 software [36] with a Bernoulli model for binary events (i.e., seropositive/seronegative). SaTScan uses Monte Carlo hypothesis testing to obtain the *p*-values and SaTScan adjusts for the underlying spatial homogeneity of a background population. For each location and size of the scanning window, the alternative hypothesis was that there was an elevated risk within the window as compared with outside and a likelihood ratio test was performed. Multiple different window sizes were used and the locations were the latitude/longitude for each animal with slight jittering to avoid more than one animal being at a location. The window with the maximum likelihood was the most likely cluster, that is, the cluster least likely to be due to chance. A *p*-value was assigned to this cluster. For this analysis, we used 9999 Monte Carlo replications, and a cluster was considered to be statistically significant if the *p*-value was <0.05.

Univariable analysis was performed using the *epiR* and *epitool* packages. The multivariable model was estimated using a Firth’s adjusted logistic regression model and the *logistf* function in the *logistf* package [37] to deal with low counts for certain covariate patterns. This was used as a guide to compare with the generalized mixed-effects model fitted using the glmmTMB package with district name as a random effect (Appendix A). Only three regions with the highest number of seropositive animals (Kilimanjaro, Tanga and Njombe regions) were included in the multivariable model. Variables with *p*-values equal or less than 0.2 in the univariable analysis were included in the multivariable analysis. Animal age and breed were also included regardless of their *p*-value due to their inherent nature; sex was not included due to the zero value in one of the 2 by2 table cell, which could have caused errors during analysis. In multivariable analysis, the *p*-value for analysis of variance for dropping each value was also recorded for the Firth’s regression model with the same variables as a check. A maximum model was fitted and variables removed in a stepwise manner checking for confounding. The final best fit was based on the lowest Bayesian information criterion (BIC). Multicollinearity was checked using the Pearson correlation test on the variable pairs implemented in the *ggpairs* function from the GGally R package.

## 3. Results

### 3.1. Brucellosis Seroprevalence in Smallholder Dairy Cattle in Selected Regions of Tanzania

A total of 2048 dairy cattle from 1374 smallholder farms were sampled from six regions of Tanzania and had complete test results (Table 1). Most animals (~64%) came from small farms with <4 animals (median herdsize = 2, IQR = 1) and there were four large farms in Iringa and Tanga with >90 animals. The cELISA was used to detect the circulating antibodies against *Brucella* species for all the samples and a total number of 49 dairy cattle were seropositive giving an unadjusted animal level seroprevalence of 2.39% (95% CI 0.017–0.31) and an adjusted animal level seroprevalence of 1.82% (95% CI 1.71–1.94) accounting for the design. Among the six regions, the Njombe Region was significantly associated with brucellosis (*p* < 0.05) and had the highest individual seroprevalence of 15.5% (95% CI 0.11–0.22) (Table 1 and Figure 2).

### 3.2. Brucellosis Hotspot Areas

The spatial choropleth map (Figure 2) shows that the seropositive animals were clustered within a small number of local authorities in the Njombe and Kilimanjaro Regions. There was a total of 29/187 seropositive animals in the Njombe Region representing 26 farms with one or more positive animals from a total of 136 farms sampled. Only three seropositives came from farms that reported an abortion in the previous 12 months and only two seropositives were cattle that the owner believed had previously had an abortion. In Kilimanjaro, there were a further 13/521 seropositive animals in the region representing 13 farms from a total of 379 farms sampled.

### 3.3. Brucellosis Spatial Clustering of Seropositive Animals

To explore this further, only the Njombe and Kilimanjaro Regions were mapped (Figure 3), and visually, there appeared to be a cluster of positive animals in the northern part of the Njombe Region and also in Kilimanjaro Region. This was more formally tested using the spatial clustering test (*p* < 0.05) which identified a cluster of 84 animals within which were all 29 seropositives with relative risk of 26.4 (95% CI 3.7–190.6) and a radius of 21.14 km in the Njombe Region (Figure 4), and a further cluster comprising 49 animals with seven positive animals with a relative risk of 11.2 (95% CI 3.9–32.1) and radius of 3.93 km was identified in the Kilimanjaro Region (Figure 3b). The other cluster had fewer seropositive animals and was not statistically significant.

### 3.4. Age Stratification of Seropositive Animals

In order to try to assess if this was a single outbreak or more of an endemic expansion, the age-stratified seroprevalences were plotted (Figure 4). The overall age-stratified seroprevalences do not suggest any strong increase with age. In fact, it is very low across all ages, with the possible exception of the animals over 8 years old. However, given the apparently clustered pattern of seropositivity, the Njombe and Kilimanjaro Regions were separated out and plotted on their own (inset Figure 4). Again, the seroprevalence appears very uniform across ages in the Njombe Region at ~18% and the Kilimanjaro Region at ~2% (red dashed line).

### 3.5. Univariable Analysis Results

The initial univariable screening (Table 2) identified several factors at the farmer level that were associated with increased likelihood of animals being brucellosis seropositive, including level of education, knowledge and experience in dairy cattle keeping, and training and years of experience keeping cattle, which passed the threshold for inclusion in the multivariable model (*p* < 0.2). Cattle sex and whether the farmer had their own bull, although potentially associated (*p* < 0.01), had too few observations in one cell to be of use in the multivariable model. Only a small number of animals were from herds reporting routine vaccination against brucellosis, and all the animals from herds reporting vaccination were seronegative. There was some evidence supporting a breed effect, but animal age and management of feeding (zero grazed or at pasture) did not have any evidence of an association (*p* > 0.05). Herd size showed a decreasing risk with increasing herd size. The source of drinking water appeared to be associated with seropositivity (*p* < 0.05), but there were small numbers in one cell and the CI were very large and included a value of one. Distance between farms was associated (*p* < 0.05) with an increased risk for farms more than 100 m apart. Contact with goats, sheep, and dogs were all strongly associated with increased risk of seropositivity (*p* < 0.05). Correct disposal of the placenta and a history of abortion in the herd were also factors associated with an increased risk (OR > 1, *p* < 0.05). Finally, the clustered spatial pattern was captured by the two potential stratifying variables for region or zone with Njombe/Southern in particular showing a very strong association with being seropositive.

### 3.6. Final Multivariable Mixed-Effects Logistic Regression Model

The final model and the backward stepwise approach to the final model is shown in Appendix A. Towards the end of the selection, there were problems with large CIs for the zone variable with the glmm as compared with the Firth’s model. This was not improved by using region and, in the end, removing Zone from the glmm and leaving the random effect for the spatial component produced a much more stable model (Appendix A). Variables were reintroduced to check for any improvement in fit, and also an interaction between sheep and goat was tested. However, the final best fitting most parsimonious model included only the contact with goats variable and whether the farm had a recent (last 12 months) history of abortion on the farm (Table 3). Animals with contact with goats either on their own or on neighboring farms had a 3.02 (95% CI 1.22–7.46) increased odds of being seropositive, and animals from farms with a recent history of abortion had a 4.91 (95% CI 1.43–16.9) increased odds of being seropositive as compared with farms with no recent history of abortion.

Interestingly, there was no association with age, consistent with the age stratification analysis (Figure 4). In addition, the intracluster correlation coefficient (ICC) was very high at 0.641, suggesting that animals within clusters (in this case administrative districts) were very highly correlated.

## 4. Discussion

Brucellosis is one of the globally, neglected bacterial zoonosis [2] that continues to pose huge economic losses in LMICs including Tanzania due to lack of effective control measures and well-established surveillance systems [8,10] In Tanzania, recent studies have shown that brucellosis in dairy cattle is re-emerging [12]. Therefore, the current study aimed to determine the status and emerging risk factors for brucellosis in smallholder dairy cattle in selected high milk-producing regions of Tanzania.

The findings indicated the presence of circulating anti-*Brucella* antibodies in dairy cattle, suggesting the presence of brucellosis in selected regions. The population-adjusted animal level seroprevalence was 1.82% (95% CI 1.71–1.94) across all study regions. The seroprevalence was highest (15.5%) in the Njombe Region, while the Mbeya Region had no seropositive animals, and the Iringa and Arusha Regions only had one animal each. A similar study by Mathew et al. (2015) [24] provided evidence that the Njombe Region could be a brucellosis hotspot under smallholder dairy systems [24]. High seroprevalences were also reported in the Morogoro, Iringa, and Tanga Regions under smallholder dairy systems [12,14,19] but this was not observed in this study. The current findings suggest that mitigation measures for controlling brucellosis in smallholder dairy cattle must be prioritized and instituted especially in high-risk areas.

Brucellosis in smallholder dairy cattle in the Kilimanjaro Region is re-emerging. This study found that the region-specific seroprevalence in smallholder dairy cattle in the Kilimanjaro Region was 2.5%. A research study carried out in Moshi-Kilimanjaro in smallholder dairy cattle in 2000 found an animal-level seroprevalence of 12.2% [25]. However, a recent study carried out in the Kilimanjaro Region and nearby regions in dairy cattle populations found an overall animal-level seroprevalence of only 0.01% [38] suggesting that the disease was on a decreasing trend. However, the seroprevalence of 2.5% found by this study in the Kilimanjaro Region suggests that brucellosis is re-emerging, and therefore, dairy farmers must strengthen control measures for brucellosis to avoid it becoming endemic.

The spatial clustering analysis results suggest that two clusters were significantly associated with brucellosis seropositivity, one in the Njombe Region and the other in the Kilimanjaro Region, and the intracluster correlation of 0.61 suggested there was potentially a localized problem. Again, the seroprevalence appears to be very uniform across ages in the Njombe and Kilimanjaro Regions. This very clustered spatial pattern along with the very uniform age seroprevalence profile are more consistent with an outbreak of brucellosis rather than with a widespread general endemicity where you might expect to see seroprevalence increasing with age [39].

In the Mbeya Region, this study found no seropositive dairy cattle, contrary to the findings of previous studies which found seroprevalence ranging from 2.8% to 17.8% [13,15,40]. The decreasing trend of brucellosis in the Mbeya Region might be attributed to the implementation of control strategies following the findings of the previous studies.

Brucellosis in smallholder dairy cattle is also affecting neighboring countries in East and Southern Africa. Studies carried out in Rwanda, Zambia, Malawi, Burundi, and Uganda have recorded higher brucellosis seroprevalence of 23.1%, 6.0%, 7.7%, 14.7%, and 6.0%, respectively [41,42,43,44,45], than the overall findings of this study in Tanzania. Countries such as Kenya and Ethiopia have recorded seroprevalence between 1% and 1.9% which is lower than the findings of this study in Tanzania [1,10,29]. This suggested that brucellosis is present in Sub–Saharan Africa, and hence, regional efforts are required for the strategic control of brucellosis.

The univariable analysis demonstrated a number of variables significantly associated with *Brucella* seropositivity that were not maintained in the final model. We would like to highlight a couple of these because of their importance to control measures for *Brucella* in animals and people.

Dairy cattle kept by farmers who had livestock training were significantly more likely to be seropositive (OR 2.03, *p* < 0.05). These farmers had attended informal livestock training by mostly having meetings with livestock field officers for a few hours a day, the trainings normally focused on production and did not cover disease control extensively. Training and education on zoonotic diseases should be a focus of future programs to reduce the risks to cattle and people. The univariable analysis demonstrated that cattle on dairy farms which kept a bull were less likely to be brucellosis seropositve (OR 0.06, *p* < 0.05); this finding was in agreement with the findings of other studies which found that keeping or owning a bull for breeding was protective [46]. The use of brucellosis-free bulls for breeding is critical to reduce chances of spreading the disease to cows. Placenta disposal was significantly associated with seropositivity after the univariable analysis; dairy cattle on a farm which practiced incorrect disposal methods, such as feeding it to dogs, throwing it away in an open dump, or just leaving it, were 4.06 times more likely to be seropositive as compared with dairy cattle on a farm which practiced correct disposal methods such as burying, burning, or putting in toilets. Other studies have found that incorrect disposal of fetal membranes increases the chance of seropositivity [47,48]. Another study found that incorrect disposal of fetal membranes was associated with poor hygiene on farms, which led to contamination of pasture and water, and eventually, infection of cattle; however, there was no statistical significance [42]. This may also pose a risk for human exposure and appropriate disposal should be taught during training sessions.

Routine vaccination against brucellosis had a zero value in one of the 2 × 2 cells, and therefore, the analysis could not be performed. In Tanzania, the S.19 vaccine for brucellosis in cattle is produced by TVLA. However, vaccination is not widely practiced, similar to other African countries [42]. In order to promote vaccination, the Tanzanian government issued a list of strategic diseases which included controlling brucellosis by vaccination. In this study, some of the farmers claimed to have vaccinated their animals against brucellosis during interviews; however, on cross examination, there was no proof of vaccination. It is known that an effective control method for brucellosis in cattle is by vaccination [49,50]; however, some studies have not found a significant association between vaccination and brucellosis seropositivity (protection) [51,52], which may be attributed to the challenges associated with vaccination coverage, as well as vaccination and vaccine handling [52].

The multivariable mixed-effects model for brucellosis seropositivity for the three regions found that dairy farms with goats and farms which had histories of abortion during the past 12 months were significantly associated with brucellosis seropositivity.

Dairy cattle kept on farms which had goats were more likely to be seropositive (OR 3.02. 95% CI 1.22–7.46) than dairy cattle on farms which did not have goats. This finding was in agreement with the findings of other research studies that reported dairy cattle kept together with goats increased the risks of cattle contracting *Brucella* infection [1,46,53]. Brucella cross-species infection has been reported, B. melitensis has been identified in dairy cattle [54,55], and in Tanzania, B. abortus has been identified in goats [56]. The epidemiology of brucellosis in cattle is becoming more complex in mixed farming systems which are common practice in Tanzania and in LMICs.

In light of mixed farming practices and the cross-species infection nature of *Brucella*, identification of *Brucella* species circulating in dairy cattle populations is becoming increasingly important rather than measuring brucellosis seropositivity of cattle. Control of brucellosis by using the monovalent *B. abortus* S19 vaccine might be redundant if *B. melitensis* infects cattle. In Sub-Saharan Africa, small ruminants have been identified as being a challenge for the control of brucellosis and the primary cause of reemergence of brucellosis in cattle [57]. In Tanzania, brucellosis studies have been focused on cattle, with few studies in small ruminants. Future work should focus on understanding the roles of small ruminants in the epidemiology of brucellosis in cattle in its wider dimensions [58].

Brucellosis causes abortion in pregnant cows, and a history of abortion has been associated with brucellosis. This study found that a history of abortion was significantly associated with brucellosis seropositivity and that dairy cattle kept in a herd with a history of abortion were more likely to be brucellosis seropositive as compared with those in herds with no history of abortion (OR 4.91, 95% CI 1.43–16.9). These findings were in agreement with the findings of other studies in dairy cattle [11,41,42]. Farmers should be encouraged to report abortions in dairy cattle to the livestock officials for closer monitoring of the disease and implementation of control measures.

## 5. Conclusions and Recommendations

The current study on the status and risk factors for brucellosis in smallholder dairy cattle in selected regions of Tanzania confirms that brucellosis in smallholder dairy cattle is still a problem, and in some regions, it is re-emerging. The Njombe Region was identified as a hotspot and, since it is a region with emerging smallholder dairy farming, there needs to be further surveillance and control programs to manage the disease. Keeping goats or having goats around dairy cattle is an emerging risk factor for brucellosis in dairy cattle. In addition, dairy farmers should be educated about the risk and mitigation measures taken to reduce transmission between species and particularly the public health risk.

## Figures and Tables

**Figure 1 vetsci-10-00155-f001:**
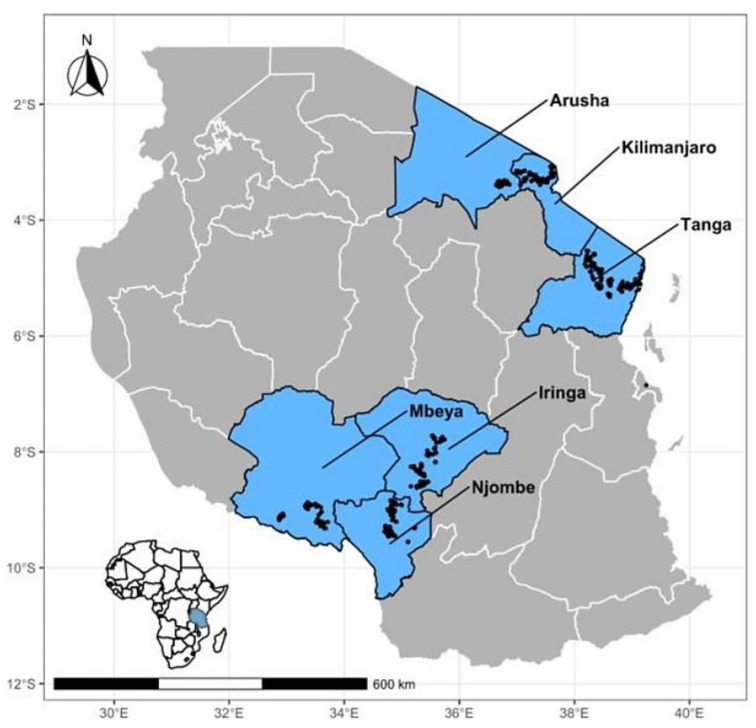
Map of Tanzania showing study regions (in light blue color) with high population of smallholder dairy cattle and unstudied regions in gray (right). Black dots indicate the locations of cattle sampled. Inset (top right corner) shows the location of Tanzania in Africa.

**Figure 2 vetsci-10-00155-f002:**
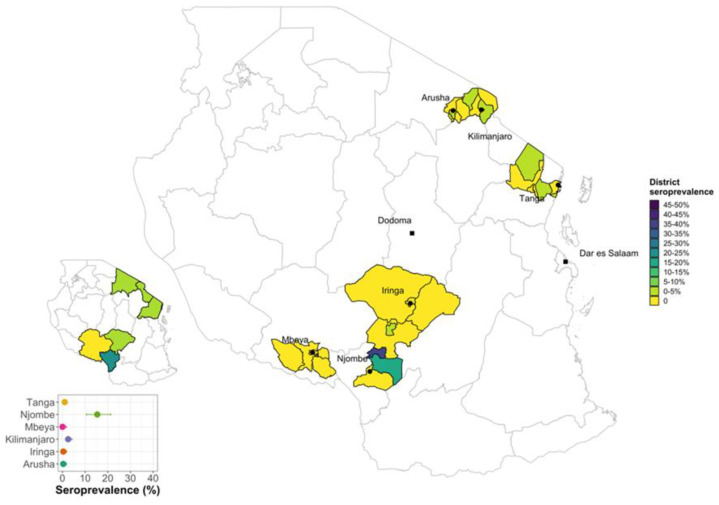
Choropleth map showing the regional seroprevalences (insets) and the detailed seroprevalence by local authority sampled in each region.

**Figure 3 vetsci-10-00155-f003:**
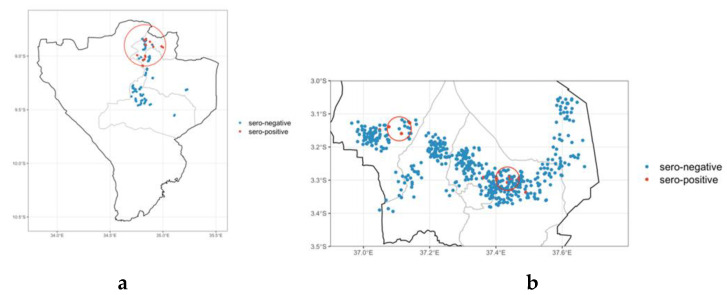
Map of Njombe (**a**) and Kilimanjaro (**b**) Regions showing district boundaries, the location of seropositive and seronegative animals (jittered), and the radius (red circle) of the significant clusters identified by the SaTScan analysis. In the Kilimanjaro Region, the top left cluster was not statistically significant.

**Figure 4 vetsci-10-00155-f004:**
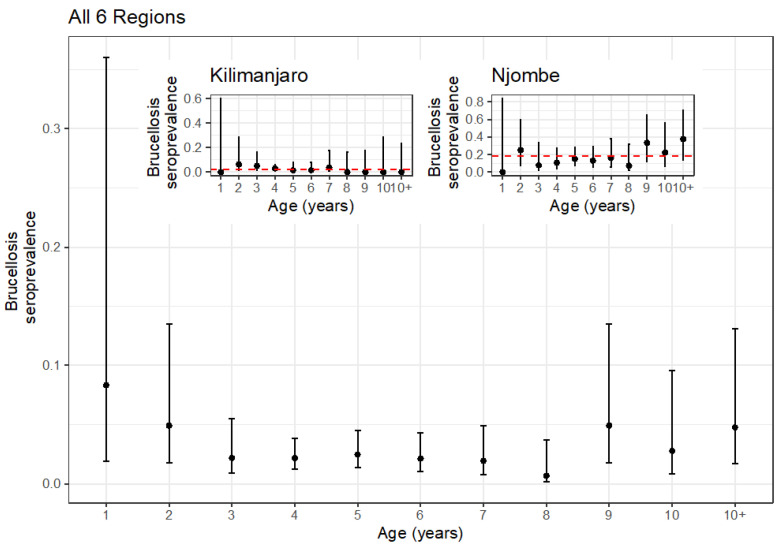
Age-stratified seroprevalence of brucellosis across all six regions. Inset: Age-stratified seroprevalence for the Kilimanjaro and Njombe Regions with a red dashed line for the mean regional seroprevalence.

**Table 1 vetsci-10-00155-t001:** Seroprevalence of brucellosis at the animal level in the study regions of Tanzania.

Animal Level Seroprevalence_cELISA							
Region	Negative	Positive	Total	Prev %	95% CI	Pop	Weights
Arusha	317	1	318	0.3	0.00–1.74	78,637	247
Kilimanjaro	508	13	521	2.5	1.65–4.2	41,639	79
Tanga	519	5	523	1.0	0.3–2.2	161,984	311
Mbeya	217	0	217	0.0	0.0–1.6	72,724	335
Iringa	280	1	281	0.4	0.0–1.9	7081	25
Njombe	158	29	187	15.5	11.0–22.0	7177	38
TOTAL	1999	49	2048	2.39	1.7–3.1	369,242	

Pop, population; Prev, prevalence; CI, confidence interval.

**Table 2 vetsci-10-00155-t002:** Univariable analysis based on the 3 regions with seropositive dairy animals for *Brucella* in Tanzania.

		Number			95% CI		
Variables	Levels	Negatives	Positives	OR	Lower	Upper	*p* Value
Farmer’s gender	Female-headed farms	537	17	1			
	Male-headed farms	648	30	1.46	0.8	2.68	0.24
Livestock training attended	No	907	29	1			
	Yes	278	18	2.03	1.11	3.7	0.02
Education level attained	Basic (none or primary only)	886	44	1			
	Secondary +	299	3	0.76	0.14	3.98	0.003
Experience in keeping dairy cattle	<5 years	58	9	1			
	≥5 years	1127	38	0.37	0.18	0.77	0.006
Cattle sex	Female	1164	47				
	Male	21	0				
Do you own Bull	No	967	46	1			
	Yes	218	1	0.06	0.01	0.41	<0.01
Do you routinely vaccinate for *Brucella*	No	1164	47				
	Yes	21	0				
Breed	Other	9	1	1			
	SHZxAyshire	258	6	0.21	0.02	10.7	
	SHZxFriesian	824	39	0.43	0.06	19.1	
	SHZxJersey	94	1	0.10	0.00	8.3	0.09
Age of cattle	<5 years	514	18	1			
	5–7 years	527	20	1.08	0.54	2.2	
	>7 years	144	9	1.78	0.69	4.3	0.351
Feeding management	Pasture	207	5	1			
	Zero-grazed	978	42	1.78	0.7	4.6	0.323
Herd size	1–2 cows	548	32	1			
	3–4 cows	402	13	0.55	0.26	1.1	
	>4 cows	235	2	0.15	0.02	0.58	0.004
Water source	River	121	2	1			
	Tap	889	26	1.77	0.43	15.57	
	Well	175	19	6.54	1.53	58.97	<0.01
Distance between herds	<100 m	923	22	1			
	≥100 m	262	25	4.0	2.22	7.22	<0.01
Dogs	No dog	775	22				
	Have dogs around	410	25	2.15	1.2	3.86	0.01
Goats	No goats	389	7	1			
	Have goats around	794	40	2.8	1.24	6.31	0.01
Sheep	No sheep	979	30	1			
	Have sheep around	202	17	2.75	1.49	5.07	<0.01
Pigs	No pigs	993	40	1			
	Have pigs around	192	7	0.91	0.40	2.05	0.811
Region	Tanga	519	5				
	Kilimanjaro	508	13	2.65	0.88	9.58	
	Njombe	158	29	18.95	7.1	63.8	<0.01
Zone	Northern	1027	18	1			
	Southern	158	29	10.47	5.68	19.3	<0.01
Placenta disposal	Correct	1102	83	1			
	Incorrect	36	11	4.06	1.99	8.26	<0.01
Abortion history (within herd)	No	1133	42	1			
	Yes	52	5	2.59	0.99	6.83	0.06

OR, odds ratio; %, percent; CI, confidence interval.

**Table 3 vetsci-10-00155-t003:** Final multivariable mixed-effects model for *Brucella* seropositivity in dairy cattle in the Tanga, Kilimanjaro and Njombe Regions of Tanzania. The intracluster correlation coefficient was 0.641.

Risk Factor	OR	95% CI
Goats		
no goats	1	-
have goats around	3.02	1.22–7.46
Abortion history		
no	1	-
yes	4.91	1.43–16.9

OR, odds ratio; CI, confidence interval.

## Data Availability

All relevant data are presented within the manuscript.

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
