# Peer review of "The Status and Risk Factors of Brucellosis in Smallholder Dairy Cattle in Selected Regions of Tanzania"

_vetsci, 2023, doi:10.3390/vetsci10020155_

Round 1

Reviewer 1 Report

The manuscript entitled "The status and risk factors of brucellosis in smallholder dairy 2 cattle in selected regions of Tanzania" was reviewed. This is a very interesting epidemiological study on a disease with zoonotic potential. 

There is one major issue that has to be addressed regarding the evaluation of seropositivity. The authors should comment and justify the choice of the analytical method they used. Morever, they should describe the diagnostic performance of the specific method so as to support the validity of their results.

The maps are perfect. However, the location of Tanzania in a map of Africa will be a nice addition for the international readership of this journal. 

Regarding the animal selection the authors refer to the "ADGG project that has registered over 52,500 cattle 111 across the regions from volunteer farmers, and approximately 4000 cattle were randomly 112 selected for genotyping as part of genetic evaluations of this crossbred population. I am wondering why the authors did not make any reference to the effect of different genotypes on seropositivity. It would be nice to have such an interesting information.

Author Response

Comment

There is one major issue that has to be addressed regarding the evaluation of seropositivity. The authors should comment and justify the choice of the analytical method they used. Moreover, they should describe the diagnostic performance of the specific method so as to support the validity of their results.

Response

World Organization for Animal Health (WOAH) recommends the use of cELISA for confirming prevalence of infection to Brucella in animals, furthermore the test has high sensitivity ( 97.9%) and specificity (~100%) and can be used for testing poor quality serum samples (Stack et al, 1999, OIE, 2018).

Comment

The maps are perfect. However, the location of Tanzania in a map of Africa will be a nice addition for the international readership of this journal. 

Response

An inset of map of Africa showing the location of Tanzania was added on top right corner of map of Tanzania.

Comment

Regarding the animal selection the authors refer to the "ADGG project that has registered over 52,500 cattle 111 across the regions from volunteer farmers, and approximately 4000 cattle were randomly 112 selected for genotyping as part of genetic evaluations of this crossbred population. I am wondering why the authors did not make any reference to the effect of different genotypes on seropositivity. It would be nice to have such an interesting information.

Response:

The ideas is worth to include but this interesting information about the effect of genotype on seropositivity and or PCR positivity has been planned to be reported separately in another paper by another researcher in our team.

Reviewer 2 Report

dear author

the review is in file

thanks

Author Response

Reviewer 2.
Line 46 - hotpots – it is need to change the word to hotspots.
Response: changed to hotspots
Line 77 – kind of test? Soroaglutination? Elisa? Pcr?
Response: I added:
by using serum agglutination test
Line 90 - highest density? What number? It is better the author should be more specific
Response: some changes on sentence: farms was deleted and cattle was added, zonal number of dairy
cattle were added as cited from the reference:
The study was conducted from July 2019 to October 2020 across six administrative regions in Tanzania
with the highest density of smallholder dairy cattle [27]..
Arusha, Kilimanjaro and Tanga regions in the Northeastern highland zone with 252,554 heads of dairy
cattle and Iringa, Njombe and Mbeya regions in the Southwestern highland zone with 103,306 heads of
dairy cattleLine 114 - Mrode et al., 2021 – the reference is not in correct reference. The author need to
put in standard.
Response:
The name of researcher and year of publication were removed to remain with standard intext citation as
per Vet. Sciences.
Line 118 - The final sample size per region was sufficient to estimate seropositivity of 5% (with 3%
precision) and 95% confidence for the smallest region assuming simple random sample (as herds were
small and mostly only animal had been genotyped per herd). The text will be better written. The
sentence is very simple
Response: The sentence was changed to read:
Due to small herd size in most farms only one animal was genotyped and therefore sampled. However,
the final sample size from each region was sufficient to estimate the seroprevalence of 5% (with 3%
precision) and 95% confidence interval for the smallest region assuming simple random sample.
Line 122 - Questionnaire administration – the link of the questionnaire need be attached.
Response: I need a link to questionnaire:
Line 196 - three regions? What the region?
Response
Only three regions with highest number of seropositive animals (Kilimanjaro, Tanga and Njombe
regions) were included in multivariable model.
Line 314 – in discussion, all the factors in table 2 need be discussed. The discussion is very simple. It is
not very discussed the text.
Response
It is not common practice to discuss univariable results since the multivariable model will account for
confounding and is a more accurate representation of the risk factors. However we have added some
discussion around factors that are important for the control of Brucella and should be targeted for
education campaigns.